# Ecological Study of Sick Building Syndrome among Healthcare Workers at Johor Primary Care Facilities

**DOI:** 10.3390/ijerph192417099

**Published:** 2022-12-19

**Authors:** Loganathan Salvaraji, Shamsul Bahari Shamsudin, Richard Avoi, Sahipudin Saupin, Lee Kim Sai, Surinah Binti Asan, Haidar Rizal Bin Toha, Mohammad Saffree Jeffree

**Affiliations:** 1Public Health Medicine Department, Faculty of Medicine and Health Sciences, Universiti Malaysia Sabah, Jalan UMS, Kota Kinabalu 88400, Sabah, Malaysia; 2Astar Laboratory Snd. Bhd., 12-02, Jalan Permas 10/5, Bandar Baru Permas Jaya, Masai 81750, Johor, Malaysia; 3Public Health Division, Johor State Health Office, Malaysia Ministry of Health, Kempas Baru, Johor Bahru 81200, Johor, Malaysia

**Keywords:** sick building syndrome, indoor air quality, healthcare workers, health clinic

## Abstract

Introduction: Persistent exposure to indoor hazards in a healthcare setting poses a risk of SBS. This study determines the prevalence of and risk factors for SBS among healthcare workers in health clinics. Methods: A cross-sectional study was conducted across four health clinics from February 2022 to May 2022. As part of the study, self-administered questionnaires were completed to determine symptoms related to SBS. An indoor air quality (IAQ) assessment was conducted four times daily for fifteen minutes at five areas in each clinic (laboratory, lobby, emergency room, pharmacy, and examination room). Result: Most of the areas illustrated poor air movement (<0.15 m/s), except for the laboratory. The total bacterial count (TBC) was above the standard limit in both the lobby and emergency room (>500 CFU/m^3^). The prevalence of SBS was 24.84% (77) among the healthcare workers at the health clinics. A significant association with SBS was noted for those working in the examination room (COR = 2.86; 95% CI = 1.31; 6.27) and those experiencing high temperature sometimes (COR = 0.25; 95% CI = 0.11; 0.55), varying temperature sometimes (COR = 0.31; 95% CI = 0.003), stuffy air sometimes (COR = 0.17; 95% CI = 0.005; 0.64), dry air sometimes (COR = 0.20; 95% CI = 0.007; 0.64), and dust sometimes (COR = 0.25; 95% CI = 0.11; 0.60) and everyday (COR = 0.34; 95% CI = 0.14; 0.81). Only healthcare workers in the examination room (AOR = 3.17; 95% CI = 1.35; 7.41) were found to have a significant risk of SBS when controlling for other variables. Conclusion: SBS is prevalent among healthcare workers at health clinics.

## 1. Introduction

Sick building syndrome (SBS) has complex mechanisms of interaction in relation to body systems and agents in a building environment. Hazards causing SBS act via four mechanisms (immunology, infectious, toxic, and allergy mechanisms) in a human body [1]. SBS can be described as a collection of general, mucosal, and eye symptoms. The progress of the disease can worsen over time, and if identified too late, it can be irreversible [2]. Genetic–environmental interactions influence the development of respiratory morbidity. Indoor air quality, especially the particulate matter 2.5 parameter (PM_2.5_), is associated with the onset of skin allergies [3]. In addition, chronic fatigue syndrome (CFS) and Gulf War syndrome (GWS) have been linked to untreated SBS [4]. SBS has also been related to psychological conditions, such as work stress, chronic fatigue syndrome, and burnout [5]. According to the International Labor Organization (ILO), indoor air pollution may increase the incidence of SBS in high-risk groups, such as babies, the elderly, and people with chronic diseases [6].

Indoor air pollutants contribute to 5% of the global burden of illness around the world. This figure is related to disability-adjusted life years (DALYs) [7]. Workers who experience SBS face serious consequences, such as productivity loss and a weakened ability to perform work activities. This results in reduced amounts of overtime worked and increased staff turnover. In addition, workplace decisions are indirectly affected by an increase in sickness and, therefore, absenteeism. Inland Revenue has reported that half of workers who are unable to attend the workplace are suffering from SBS [8]. In England, this results in organizational losses of productivity due to sick leave (3%) and workers performing at reduced productivity (33%). A research study conducted in the US estimated that SBS could cost USD 5 billion. The overall cost may equal 0.5–0.1% of the nation’s gross domestic product after accounting for additional expenses for healthcare, social security, and building maintenance [9].

Healthcare personnel are faced with a vast number of risk factors related to occupational health and safety. Risks in an indoor working environment are related to heating; cooling; and chemical, biological, and work activities [10]. In the healthcare setting, biological and chemical hazards are the main concerns related to the homeostasis of workers’ bodies. Although water treatment and the chlorination of healthcare distribution systems are carried out, hazards can be transported at variable concentrations via building infrastructure, such as ventilation, furniture, and water systems [11].

Inadequate reviews and investigations into SBS and indoor air quality lead to undetermined primary sources of origin at health clinics. Furniture, machines, and carpets contain chemicals that evaporate into the air and circulate in ventilation systems. Chemicals, such as volatile organic compounds, formaldehyde, and dust, pose threats to acute conditions [12]. A combination of materials in the building can affect the perceived indoor air quality. For instance, the combination of oriented strand boards (OSBs) and painted gypsum boards may elevate the concentration of TVOC in the indoor environment [13].

Any defect in ventilation systems can lead to the poor circulation of air containing pollutants, such as chemicals, dust, and microorganisms. Inadequate material selection and poor renovation practices result in unsafe air quality for occupants’ health. Upon the completion of building renovation, SBS can last for 3 months in occupants, particularly in confined spaces [14,15]. These health symptoms, in turn, influence their ability to undertake the tasks required of them [6]. An assessment at an ambulatory care center highlighted that healthcare personnel complain of discomfort and coldness due to reduced temperatures and poor air ventilation systems, especially during non-peak hours [16]. A study at a Nigerian hospital highlighted the different satisfaction levels of the clients and suggested necessary measures to improve the indoor conditions of the building [17]. 

IAQ standards for buildings are accessible; however, specific suitable guidelines for primary care facilities are still unclear [18]. Although every workplace has an understanding of SBS, the awareness of SBS is still poor among workers in Malaysia [19]. A specific training module for SBS that provides comprehensive information on SBS and SBS-related risk factors has not yet been developed. The surveillance data of SBS and indoor air quality in health clinics are not recorded in most primary care settings. Hence, statistical projections cannot be analyzed or compared for evidence-based decisions and amendments to occupational safety and health policies. Over the years, several organizations have developed monitoring and control systems for buildings, such as IAQ sensors, Internet of Things (IoT) approaches, wireless sensor network (WSN)-based systems, air purification technology, and smart home technology [18]. However, the effectiveness of these monitoring systems is still disputed, as well as costly.

Scant research studies in the field of SBS and indoor air quality at health clinics are available worldwide. The current articles related to SBS emphasize hospital settings and ambulatory care. Primary care health facilities share similar environments for job tasks related to medical examination, medical procedures, laboratory investigation, and dispensaries. In addition, the advancement of building materials had led to uncertain air quality in relation to the global climate and land use [20]. Therefore, the objective of this study is to determine the prevalence and risk factors of SBS among healthcare workers in health clinics.

## 2. Materials and Methods

### 2.1. Subject Recruitment and the Selection of Health Clinics

This was a cross-sectional study conducted across 3 health clinics in Pontian District and 1 health clinic in Johor Bahru District. The selection of the health clinics was based on their centralized ventilation systems (air handling unit (AHU)). Each room was installed with an air supply and an outlet to remove the air from the area (Figure 1). The ventilation system operated during weekdays from 8 o’clock in the morning to 5 o’clock in the evening. Data were collected from February 2022 to May 2022. Healthcare workers who had been working for at least 4 months at the selected health clinics were invited to participate in this study. However, those who were pregnant or diagnosed with a chronic respiratory disease were excluded. Consent was obtained before participation in this study.

### 2.2. The Prevalence of Sick Building Syndrome (SBS)-Related Symptoms

A standardized questionnaire related to SBS was adapted from the Malaysian Industry Code of Practice (ICOP) on Indoor Air Quality 2010 [21]. The SBS symptom assessment was divided into 3 groups: general, mucosal, and skin symptoms. General symptoms included headache, dizziness, fatigue or lethargy, and nausea or vomiting. Mucosal symptoms focused on eye irritation, swollen eyelids, swollen eyelids, runny nose or nasal catarrh, nasal obstruction or blockage, throat dryness, sore throat, and irritating cough. Rashes on the hands or forearms, rashes on the face or throat, eczema, itchiness of the face or throat, and itchiness on the hands or forearms were categorized as skin problems. Workers who had at least one SBS symptom that occurred every day or one to four times per week in relation to the working environment and workers who had symptoms that were resolved after leaving the building, during the weekend, or on days off were coded as “1”. Meanwhile, the other participants were coded as “0” [3].

### 2.3. IAQ Measurement and Assessment

For indoor air quality sampling, five areas (laboratory, lobby, emergency room, pharmacy, and examination room) in the clinic were selected for the indoor air quality (IAQ) measurement. These areas have a high number of occupants and frequent contact with patients. Real-time IAQ monitoring was conducted for one day at each clinic. The sampling was taken four times daily from the selected areas (0800–1000; 1000–1200; 1300–1500; 1500–1700). Sampling lasted for 20 min in each area and was recorded once the reading had stabilized. The sampling points were decided according to the references specified by the Malaysian Industry Code of Practice on Indoor Air Quality (MICOP) by the Department of Occupational Safety and Health, Malaysia, 2010 [21]. The required number of sampling points for a room size of 500 m^2^ is a minimum of one. The assessment included physical, chemical, and biological parameters. All measurements were conducted as specified in MICOP 2010. The physical parameters measured were air temperature, relative humidity, and air movement. Meanwhile, the chemical contaminants included carbon dioxide (CO_2_) and particulate matter (PM_10_). The total counts of bacteria and fungus were among the biological parameters measured during the assessments. Field data log sheets were used to record every reading at each point before being transferred to Excel format. The instruments used included a TSI Q-Track IAQ (Model 7565), a Testo 425 thermal anemometer, SKY2000-M4 multi-gas detectors, an Aeroqual Portable Particulate Monitor Series 500, and a Buck Bio-Culture sampling pump. The SKY2000-M4 multi-gas detectors were applied to detect ozone (O_3_), carbon monoxide (CO), formaldehyde (CH_2_O), volatile organic compounds (VOCs), and carbon dioxide (CO_2_). The O_3_ (<0.01), CO (<0.01 ppm), CH_2_O (<0.01 ppm), and VOC (<0.01 ppm) levels were below the standard levels in all areas in the clinic, and, hence, they are not discussed further. Meanwhile, the Buck Bio-Culture sampling pump was set at a flow rate of 100 L per minute (LPM) for 2 min. Trypticase soy agar (TSA) for the bacterial culture and malt extract agar (MEA) for the fungal culture were installed and sealed before the samples were transported to the laboratory. Trypticase soy agar (TSA) was incubated at 35 °C for 48 h; meanwhile, malt extract agar (MEA) was incubated at 25 °C for 5 days. All the instruments used were calibrated according to the manufacturers’ specifications, and a workplace environment assessment was included in the questionnaire set.

### 2.4. Risk Assessment Criteria and Data Analysis

The data were analyzed using the IBM Statistical Package for Social Sciences (SPSS), version 21.0. The categorical dataset was analyzed using the chi-square method or Fisher’s exact test; meanwhile, the continuous variable was tested using an independent *t*-test. A *p*-value of <0.05 was considered significant. Then, from the univariate regression analysis, variables with a *p*-value less than 0.2 were selected for multivariable logistic regression to identify the risk factor for SBS. The backward and forward stepwise approach was applied during the analysis. Significant data were identified by a p-value less than 0.05, with an adjusted odds ratio and 95% confidence interval. The incidence of SBS was further discussed in relation to the workstation environment and the IAQ measurements obtained from each area at the health clinics.

## 3. Results

### 3.1. Respondent Profile

A total of 310 healthcare workers from four health clinics participated in this study. Most of them were female (79.35%), with a mean age of 35 years (Table 1). The participants’ ethnicities were Malay (77.42%), Chinese (7.42%), Indian (6.77%), Native Sabah (6.13%), and Native Sarawak (2.26%). A small number of them had a secondary school education (21.61%), and the remainder had completed undergraduate and postgraduate studies. Most of the participants were nurses (30.97%), followed by dental officers (19.68%), health assistants (14.84%), medical officers (11.29%), medical assistants (6.45%), laboratory technicians (3.55%), physiotherapists (1.29%), and radiographers (1.29%).

### 3.2. Comparison of Indoor Air Quality Parameters at Different Health Clinic Sites

The examination room (25.58 °C (±1.08)) was the area with the highest temperature (Table 2). Most of areas in the health clinics had poor air movement of less than 0.15 m/s, especially the examination room (0.10 (±0.03)), followed by the emergency room (0.11 (±0.02)), lobby (0.12 (±0.02)), and pharmacy (0.13 (±0.02)). Meanwhile, the total bacterial count was beyond the standard limit (<500 CFU/m^3^) in the lobby (620.66 (±214.84) and emergency room (567.42 (±252.85)). Other IAQ parameters, including the humidity, carbon dioxide (CO_2_), particulate matter 10 (PM_10_), and total fungal count, were found to be at an acceptable level.

### 3.3. Prevalence and Risk Factors of Sick Building Syndrome (SBS)

The prevalence rate of SBS among the healthcare workers in the health clinics was 24.48% (77) (95% CI = 0.20; 0.29). A higher prevalence was reported in the pharmacy (42.42%), followed by in the laboratory (36.36%), lobby (26.76%), emergency room (20.83%), and examination room (20.47%) (Table 3). However, there were no associations of SBS between the areas investigated for indoor air quality. There was an association between exposure to computers, drought, high temperature, varying temperature, low temperature, stuffy air, dry air, unpleasant odor, and dust with SBS among the healthcare workers at the health clinics. During the univariate logistic regression analysis, a significant association with SBS was noted with those working in the examination room (COR = 2.86; 95% CI = 1.31; 6.27) and those experiencing high temperature sometimes (COR = 0.25; 95% CI = 0.11; 0.55), varying temperature sometimes (COR = 0.31; 95% CI = 0.003), stuffy air sometimes (COR = 0.17; 95% CI = 0.005; 0.64), dry air sometimes (COR = 0.20; 95% CI = 0.007; 0,64), and dust sometimes (COR = 0.25; 95% CI = 0.11; 0.60) and everyday (COR = 0.34; 95% CI = 0.14; 0.81) (Table 4). Only healthcare workers in the examination room (AOR = 3.17; 95% CI = 1.35; 7.41) had a significant risk of SBS when controlling for the other variables.

## 4. Discussion

### 4.1. Comparison of Indoor Air Quality Parameters at Different Health Clinic Sites

Health clinics are at risk of contaminated air, with the attendance of sick people and crowding inside the building impacting the quality of the air. Only the examination room was close to reaching the upper standard limit of temperature in this study. Human bodies exchange heat via the heat conduction mechanism for thermal balance. The metabolic heat produced by the body is removed by 30% convection, 40% radiation, 20% evaporation, and 10% respiration [22]. Air movement was an issue in most of the areas in the study. In an enclosed building, the human body will prevent air flow and change the original air flow mechanism. Moreover, in crowded places, the influence of the crowd may significantly worsen indoor air movement. In addition, depending on the location of the input and outlet, the degree of activity, and the size of the windows, the ventilation system may have an impact on the internal air flow [23]. Bacteria are naturally present inside buildings, and common building-associated bacteria are the saprophytic bacteria of the normal human skin, mouth, and nose. The components of bacterial structures emitted into the air include bacterial cells, bacterial spores, peptidoglycans, microbial volatile organic compounds, exotoxins, and other bacteria-growing metabolites [24]. Hence, the total bacterial count was noted to be higher in the lobby and emergency room. These study areas were congested with patients waiting for medical attention.

### 4.2. Sick Building Syndrome (SBS) Symptoms among Healthcare Workers

Teaching healthcare facilities in Iran noted a particularly high prevalence of SBS of 86.4%, greater than the value determined in this study [25]. Similarly, a high prevalence of SBS was reported at a Taiwanese medical center, where 84% of healthcare workers experienced at least one SBS-related symptom [26]. A higher prevalence of SBS was also noted among healthcare workers in Sivas, Turkey, between the range of 64.7% and 74.1% [27]. However, a study among healthcare workers in Slovenia reported a lower SBS prevalence of 12% for at least six symptoms, in addition to a 19% prevalence of SBS for at least two to three symptoms of SBS [28]. However, a study in Spain found that the prevalence of SBS among healthcare professionals was 20% [29]. In Malaysia, a study conducted across two healthcare facilities showed disparities in the results. The prevalence of SBS was higher at a healthcare facility in Selangor (38.9%) than in this study, and a hospital in Pahang reported a lower prevalence of SBS, at 7.5%, among their healthcare workers [30].

### 4.3. Risk Factors for Sick Building Syndrome

In this study, only the respondents working with computers had a significant association with SBS. Another study described photocopiers, printers, or fax machine usage for more than 1 h per day as posing a more prominent risk factor among office workers [3]. These findings parallel a few studies that highlight the significant association between working conditions, such as temperature, dust, and unpleasant odor in the workplace, with SBS. In a study with the application of the Lindeman, Merenda, and Gold (LMG) test, dust and dirt, as well as stuffy “bad” air, were the predominant risk factors for SBS [31]. Meanwhile, a study in a Turkish hospital demonstrated a 1.82 times increased risk of skin symptoms with dry indoor air and a 2.17 times increased risk of non-specific symptoms [32]. An SBS score survey highlighted the presence of odor, new wall paint, the presence of fungus/mold on the walls, and the presence of a rotting/mold smell as being associated with SBS [33]. A study with 177 healthcare workers reported that working environments with dust have 2.8 times the risk of SBS; meanwhile, stuffy bad odor, dry air, and unpleasant odor increase the risk of developing SBS by 2.6 times [34].

Varying temperatures are described to have a significant association with SBS [31]. Healthcare workers have a 4.31 times higher risk of complaining of SBS with varying temperatures at healthcare facilities [27]. The respondents in this study experienced SBS when the temperature level approached the higher limit of the examination room (23–26 °C). Similarly, a study of Iranian healthcare facilities identified a significant association between high temperature and SBS [35]. The air movement reading was below the standard limit in all areas, except for in the laboratory. Stagnant airflow can increase the occurrence of SBS by 1.82 times among healthcare workers [32]. Hence, the respondents in this study experienced SBS, as most of the studied areas recorded poor air movement. The examination room in particular reported the lowest air movement reading (0.10 m/s) and 3.17 times the risk of SBS.

In this study, the maximum carbon dioxide (CO_2_) concentration (789.75 ppm) was higher than the carbon dioxide (CO_2_) concentration reported in a healthcare setting in Taiwan (700 ppm) [36]. Although carbon dioxide (CO_2_) was below the threshold level in this study, an increase in every unit of carbon dioxide (CO_2_) concentration leads to a 2.1 times higher risk of SBS among healthcare workers [34]. However, the particulate matter 10 (PM_10_) reading was at an acceptable level, but few studies have illustrated that chemical parameters, such as carbon dioxide (CO_2_) and particulate matter (PM), tend to be significantly related to SBS-related symptoms [26]. 

This study shows a high level of bacterial count in the lobby and emergency room, with SBS prevalence values of 26.76% and 20.83%, respectively. Another study of 126 healthcare workers found a positive relationship between fungal and bacterial counts with SBS, but none of the relationships were statistically significant [26]. In a review of several studies concerning healthcare facilities, it was found that droplets may potentially carry pathogens and increase the risk of respiratory infections [37]. The poor maintenance of the heating, ventilation, and air conditioning (HVAC) systems in healthcare facilities may cause inadequate ventilation and aggregate the growth of microorganisms [38]. Additionally, a survey study among healthcare workers resulted in the aggravation of infectious diseases, such as asthma, allergies, and neurological diseases, when exposed to bio-aerosols [39].

### 4.4. Limitations and Recommendations

Although primary care facilities share a similar environment to hospital settings, there are insufficient references that highlight guidelines for SBS and IAQ for the government and private sectors. Nevertheless, this study highlights the inadequate maintenance of air handling units, leading to poor indoor air quality and SBS. The studied chemical parameters (ozone (O_3_), carbon monoxide (CO), carbon dioxide (CO_2_), volatile organic compound (VOC), and particulate matter (PM)) were found to be within the standard limits. Notwithstanding, physical parameters (temperature (°C) and air movement (AM)) were found to be unsatisfactory in relation to the threshold level, which concurs with healthcare workers’ feedback on the poor air quality at their workstations. Subsequently, an inefficient indoor air environment enhances microorganism growth in particular areas (e.g., in the lobby and emergency room).

IAQ parameters should be regularly monitored at health clinics. IAQ surveys that include physical and chemical parameters can be conducted monthly or every two weeks to record surveillance data. Biological evaluation is a tedious procedure for culturing samples and avoiding cross-contamination. Therefore, the collection of biological samples might depend on results from the physical parameters, chemical parameters, and occupants’ complaints. The sampling of microorganisms should be conducted every 3 months if the physical parameter readings are abnormal in the monthly surveys. A building condition survey for health clinics is advisable, including checking the water supply, piping leaks, ventilation performance, painting, and windows. Training and education are important to continuously deliver prevention measures for SBS hazards at the workplace. Conducting training in relation to other potential occupational health and safety issues is encouraged. The management team can offer financial support and assistance to carry out this training regularly. The welfare and health conditions of workers in health clinics need to be taken care of to prevent serious detrimental health effects. Workplace history and conditions can provide information on the current control measures. Feedback from workers can be channeled to the designated team to improve the preventive measures of the health clinic infrastructure. For future studies, monitoring IAQ for several days is recommended because of the fluctuations in patient attendance at health clinics.

## 5. Conclusions

In conclusion, SBS is prevalent among healthcare workers at health clinics. Failure to maintain efficient AHU creates a poor indoor environment that poses a health risk to healthcare workers, particularly those in the examination room. Crowding inside the building increases the air pollution level and changes the quality of the air. Drought, fluctuating temperatures, stuffy air, dry air, unpleasant odor, and dust are the most common complaints about poor air quality. As a result, continuously monitoring physical and biological IAQ parameters in health clinics is critical to ensure a safe indoor environment for the occupants. It is important to conduct a risk assessment and to implement control measures in order to reduce exposure to SBS hazards. Establishing a preventive strategy depends on several variables, including managerial support, funding, engineering intervention, and policy. Periodic indoor air quality evaluations and medical surveillance should be conducted for the early screening and detection of SBS. To protect health clinic employees from SBS, standard operating procedures must be developed. Until the above preventive measures are adopted and adapted in practice, SBS will continue to pose an unseen threat to healthcare workers in health clinics.

## Figures and Tables

**Figure 1 ijerph-19-17099-f001:**
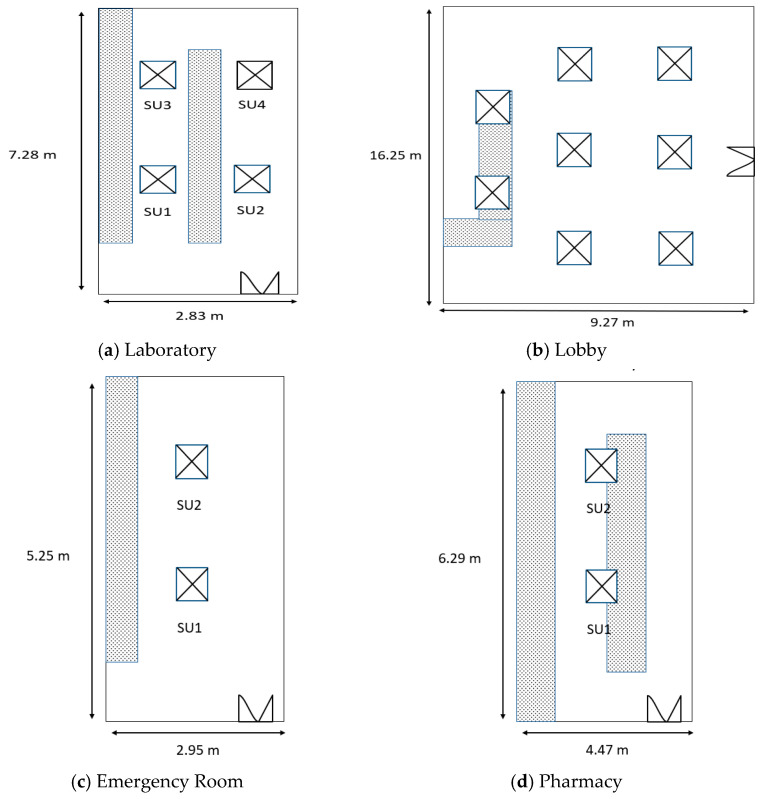
Health clinic floor plan. (**a**) room volume (area(m^2^) × height) = 7.28 m × 2.83 m × 3.00 m = 61.81 m^3^, (**b**) room volume (area (m^2^) × height) = 16.25 m × 9.27 m × 3.00 m = 451.91 m^3^, (**c**) room volume (area (m^2^) × height) = 5.25 m × 2.95 m × 3.00 m = 46.46 m^3^, (**d**) room volume (area (m^2^) × height) = 6.29 m × 4.47 m × 3.00 m = 84.35 m^3^, (**e**) room volume (area (m^2^) × height) = 4.56 m × 3.17 m × 3.00 m = 43.37 m^3^.

**Table 1 ijerph-19-17099-t001:** Respondent profiles of healthcare workers who participated in the study (N = 310).

Variable	n (%)	Mean (SD)
**Gender**		
Male	64 (20.65%)	-
Female	246 (79.35%)	-
**Age**	-	35 (±7.22)
Profession		
Medical officer	35 (11.29%)	-
Medical assistant	20 (6.45%)	-
Nurse	96 (30.97%)	-
Pharmacist	33 (10.64%)	-
Laboratory technician	11 (3.55%)	-
Physiotherapist	4 (1.29%)	-
Health assistant	46 (14.84%)	-
Radiographer	4 (1.29%)	-
Dental officer	61 (19.68%)	-
**Duration of Service**	-	7 (±6.18)

**Table 2 ijerph-19-17099-t002:** Indoor air quality parameters at different sites of the health clinics.

Location	T (°C)	RH (%)	AM (m/s)	CO_2_ (ppm)	PM_10_ (mg/m^3^)	TBC (CFU/m^3^)	TFC (CFU/m^3^)
	Mean (SD)						
Lab.	23.06 (±0.87)	57.95 (±2.59)	0.15 (±0.04)	588.30 (±35.38)	0.0059 (±0.003)	245.25 (±89.16)	151.74 (±34.50)
Lobby	24.29 (±1.13)	64.26 (±4.23)	**0.12 (±0.02)**	645.81 (±55.63)	0.0069 (±0.003)	**620.66 (±214.84)**	236.39 (±65.33)
ER	24.51 (±1.44)	59.85 (±4.89)	**0.11 (±0.02)**	798.75 (±177.35)	0.0131 (±0.017)	**567.42 (±252.85)**	199.40 (±53.24)
Pharmacy	23.50 (±0.57)	60.67 (±3.20)	**0.13 (±0.02)**	647.90 (±70.30)	0.0450 (±0.071)	255.87 (±29.21)	162.65 (±54.62)
Exam.R	**25.58 (±1.08)**	62.33 (±3.63)	**0.10 (±0.03)**	639.76 (±76.44)	0.0077 (±0.005)	281.58 (±12.67)	191.85 (±74.25)

Lab. = laboratory; ER = emergency room; Exam.R = examination room. Acceptable limit values based on ICOP 2010: T = temperature (23–26 °C); RH = relative humidity (40–70%); AM = air movement (0.15–0.5 m/s); CO_2_ = carbon dioxide (1000 ppm); PM_10_ = particulate matter 10 (<0.15 mg/m^3^); TBC = total bacteria count (<500 CFU/m^3^); TFC = total fungal count (<1000 CFU/m^3^) [21].

**Table 3 ijerph-19-17099-t003:** Univariate analysis of SBS among healthcare workers in health clinics.

Variable	SBS		*X* ^2^	*p*-Value
	Yes (n = 77)	No (n = 233)		
**Working Area**				
Laboratory	4 (36.36%)	7 (63.64%)	8.346	0.08
Lobby	19 (26.76%)	52 (73.24%)		
Emergency room	5 (20.83%)	19 (79.17%)		
Pharmacy	14 (42.42%)	19 (57.58%)		
Examination room	35 (20.47%)	136 (79.53%)		
**Equipment**				
*Computer*				
Never	7 (15.92%)	37 (84.09%)	5.983	**0.05**
2–3 times a week	15 (18.52%)	66 (81.48%)		
Everyday	55 (29.73%)	130(70.27%)		
*Photostat*				
Never	32 (22.70%)	109 (77.30%)	1.607	0.448
2–3 times a week	25 (24.04%)	79(75.96%)		
Everyday	20 (30.77%)	45(69.23%)		
*Fax*				
Never	70 (25.18%)	208 (74.82%)	1.832	0.400
2–3 times a week	4(36.36%)	7 (63.64%)		
Everyday	3(15.00%)	17(85.00%)		
**Items at workstation**				
*Carpet*				
Yes	1 (5.88%)	16 (94.12%)	3.462 *	0.082
No	76 (25.94%)	217 (74.06%)		
*New Furniture*				
Yes	-	7 (100.00%)	2.367 *	0.199
No	77 (25.41%)	226 (74.59%)		
*Recent painting*				
Yes	2 (10.53%)	17 (89.47%)	2.221 *	0.175
No	75 (25.77%)	216 (74.23%)		
*Pipe leak*				
Yes	13 (34.21%)	25(65.79%)	2.038	0.153
No	64 (23.53%)	208 (76.47%)		
*Air con*				
Yes	61 (24.70%)	186 (75.30%)	0.013	0.909
No	16 (25.40%)	47 (74.60%)		
*Refresher*				
Yes	12 (20.69%)	46 (79.31%)	0.658	0.417
No	65 (25.79%)	187 (74.21%)		
*Repellent*				
Yes	6 (26.09%)	17 (73.91%)	0.021	0.886
No	71 (24.74%)	216 (75.26%)		
**Condition at workplace**				
*Drought*				
Never	55 (21.57%)	200 (78.43%)	8.346	**0.015**
Sometimes	18 (39.13%)	28 (60.87%)		
Every day	4 (4.44%)	5 (55.56%)		
*High temperature*				
Never	23 (15.13%)	129 (84.87%)	16.453	**0.000**
Sometimes	39 (31.97%)	83 (68.03%)		
Every day	15 (41.67%)	21 (58.33%)		
*Varying temperature*				
Never	23 (17.69%)	107 (82.31%)	9.367	**0.009**
Sometimes	38 (26.95%)	103 (73.05%)		
Every day	16 (41.03%)	23 (58.97%)		
*Low temperature*				
Never	35 (19.23%)	147 (80.77%)	7.428	**0.024**
Sometimes	37 (32.74%)	76 (67.26%)		
Every day	5 (33.33%)	10 (66.67%)		
*Stuff air*				
Never	48 (20.69%)	184 (79.31%)	11.702	**0.003**
Sometimes	23 (33.82%)	45 (66.18%)		
Every day	6 (60.00%)	4 (40.00%)		
*Dry air*				
Never	43 (19.20%)	181 (80.80%)	15.449	**0.000**
Sometimes	27 (36.99%)	46 (63.01%)		
Every day	7 (53.85%)	6 (46.15%)		
*Unpleasant odor*				
Never	42 (23.46%)	137 (76.54%)	3.573	0.168
Sometimes	30 (24.79%)	91 (75.21%)		
Every day	5 (50.00%)	5 (50.00%)		
*Dust*				
Never	31 (20.13%)	123 (79.87%)	10.667	**0.005**
Sometimes	33 (25.38%)	97 (74.62%)		
Every day	13 (50.00%)	13 (50.00%)		

* Fisher’s exact test.

**Table 4 ijerph-19-17099-t004:** Univariate and multivariate regression analysis of SBS among healthcare workers in health clinics.

Variable	COR (95% CI)	*p*-Value	AOR (95% CI)	*p*-Value
**Working Area**				
Laboratory *	1.00		1.00	
Lobby	2.22 (0.615; 8.01)	0.223	3.93 (1.00; 15.49)	0.051
Emergency room	1.42 (0.75; 2.70)	0.286	1.64 (0.82; 3.28)	0.165
Pharmacy	1.02 (0.36; 2.93)	0.967	1.10 (0.34; 3.58)	0.873
Examination room	2.86 (1.31; 6.27)	**0.009**	3.17 (1.35; 7.41)	**0.008**
**Equipment**				
*Computer*				
No *	1.00		-	-
2–3 times/week	0.45 (0.19; 1.06)	0.069	-	-
everyday	0.54 (0.28; 1.02)	0.058	-	-
**Items at workstation**				
*Carpet*				
No *	1.00		-	-
Yes	5.60 (0.73; 42.97)	0.097	-	-
*Recent painting*				
No *	1.00		-	-
Yes	2.95 (0.67; 13.08)	0.154	-	-
*Pipe leak*				
No *	1.00		-	-
Yes	0.59 (0.29; 1.22)	0.157	-	-
**Condition at workplace**				
*Drought*				
Never *	1.00		-	-
Sometimes	0.34 (0.09; 1.32)	0.121	-	-
Everyday	0.80 (0.19; 3.40)	0.766	-	-
*High temperature*				
Never *	1.00		1.00	-
Sometimes	0.25 (0.11; 0.55)	**0.001**	0.37 (0.12; 1.15)	0.086
Everyday	0.66 (0.31; 1.41)	0.283	0.88 (0.31; 2.48)	0.814
*Varying temperature*				
Never *	1.00		1.00	-
Sometimes	0.31 (0.14; 0.68)	**0.003**	1.07 (0.34; 3.39)	0.915
Everyday	0.53 (0.25; 1.11)	0.092	0.92 (0.33; 2.59)	0.870
*Low temperature*				
Never *	1.00		-	-
Sometimes	0.48 (0.15; 1.48)	0.200	-	-
Everyday	0.974 (0.31; 3.05)	0.964	-	-
*Stuffy air*				
Never *	1.00		1.00	-
Sometimes	0.17 (0.05; 0.64)	**0.009**	0.47 (0.06; 3.61)	0.464
Everyday	0.34 (0.09; 1.33)	0.121	0.37 (0.05; 2.89)	0.346
*Dry air*				
Never *	1.00		1	-
Sometimes	0.20 (0.07; 0.64)	**0.006**	0.43 (0.08; 2.46)	0.345
Everyday	0.50 (0.15; 1.65)	0.258	0.99 (0.17; 5.79)	0.988
*Unpleasant odor*				
Never *	1.00		-	-
Sometimes	0.31 (0.09; 1.11)	0.072	-	-
Everyday	0.33 (0.09; 1.22)	0.096	-	-
*Dust*				
Never *	1.00		1.00	
Sometimes	0.25 (0.11; 0.60)	**0.002**	0.53 (0.18; 1.59)	0.258
Everyday	0.34 (0.14; 0.81)	**0.014**	0.46 (0.17; 1.28)	0.136

* Reference group.

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
