# Peer review of "Ecological Study of Sick Building Syndrome among Healthcare Workers at Johor Primary Care Facilities"

_ijerph, 2022, doi:10.3390/ijerph192417099_

Round 1
Reviewer 1 Report
The subject of the study is important thus there is need to improve working conditions of health care workers. The authors have selected a general term of SBS to cover the unfort and symptoms of the working conditions. It does not fit the best to describe the possible problems in particulate buildings and the details are hidden uder this common term showing if there are problems or not. The information is hidden under this common term.
The absract is descriptive as such giving the information to hte readers about the study and the results.
The measurement were done with continuous monitors. Please describe more details what parameters were really measured. T, RH and CO2 are clear, but e.g. indoor pollutants measured by SKY2000-M4 need to describe in more details, what agents were measured, and what was the range of detection for these. Have you compared the readings given by the methods based on laboratory sampling and analysis? Or have you checked this from literature?
Sampling of the bioaerosols with Buck Bio-Culture sampling pump does not describe the analysis with details needed. Please describe what media (on plates) you used for bacteria and moulds, what was incubation temperature and time.
Air velocity is an important factor for thermal comfort, but it does not give information on air quality. Air exchange rate would be more informative to show the adequancy of ventilation. However, by measuring CO2 you get the indications for this. Did you measure the ventilation rates, and did you investicate the air distribution in the different buildings. Did they differ?
Analysis of the questionnaire results was sufficiently done, although presentation was long as such.
Table 4. New furniture shows huge number of COR. Is this correct and does it mean that new furniture showed SBS symptoms or not.
In general use of terminology and language shall be check, there were some minor typos also.
Conclusions
SBS is a general term and it does not give tools as such for corrective measures in the buildings if occupants reports diffent kind of symptoms. Please present in more details based on your findings which symptoms and what reasons were the most important to cause SBS label for the building. Without these details you can't indicate the sources or reasons for symptoms that need to be improved.
Author Response
Thank you reviewer for the enlightening comments. We had tried our best to edit the article with recommended information and fruitful suggestion. We also deeply apologize for any inconvenience while preparing this article. For us it has been an really a challenging task to highlight the IAQ problems in primary care facilities. This is because of less number to healthcare workers in primary care facilities to complement sufficient sample size. In addition, IAQ instruments to be implemented in more areas or health clinics is costly. Its too complicated whereby, less number of studies conducted in primary care facilities. COVID 19 pandemic is an eye opener to emphasis the important of IAQ at primary care facilities. We tried our best to concurrent with recommended methodology to avoid biases and errors. Thank you once again.
Point 1: The subject of the study is important thus there is need to improve working conditions of health care workers. The authors have selected a general term of SBS to cover the unfort and symptoms of the working conditions. It does not fit the best to describe the possible problems in particulate buildings and the details are hidden under this common term showing if there are problems or not. The information is hidden under this common term.
Response 1: Added information
“Nevertheless, this study highlights inadequate maintenance of Air Handling Unit leading to poor indoor air quality and SBS. The chemical parameters (ozone (O3), carbon monoxide (CO), carbon dioxide (CO2), volatile organic compound (VOC), particulate matters (PM)) resulted within the standard limits. Notwithstanding, physical parameters (temperature (oC) and air movement (AM)) reported unsatisfactory than the threshold level in which concurrent with healthcare workers feedback of poor air quality at their workstation. Subsequently, inefficient indoor air environment enhances microorganism growth in particular area (lobby and emergency room).” (Line 336).
“Failure to maintain efficient AHU invent poor indoor environment that posses health risk for the healthcare workers.” (Line 367).
Point 2: The abstract is descriptive as such giving the information to the readers about the study and the results.
Response 2: Thank you for the comments.
Point 3: The measurement were done with continuous monitors. Please describe more details what parameters were really measured. T, RH and CO2 are clear, but e.g. indoor pollutants measured by SKY2000-M4 need to describe in more details, what agents were measured, and what was the range of detection for these. Have you compared the readings given by the methods based on laboratory sampling and analysis? Or have you checked this from literature?
Response 3: Added as per comment
“SKY2000-M4 Multi-Gas Detectors applied to detect ozone (O3), Carbon Monoxide (CO), Formaldehyde (CH2O), Volatile Organic Compound (VOC), Carbon Dioxide (CO2). The O3 (<0.01), CO (<0.01ppm), CH2O (<0.01ppm) and VOC (<0.01ppm) was below the standard level in all areas in the clinic, hence it was not discussed further.” (Line 162).
Point 4: Sampling of the bioaerosols with Buck Bio-Culture sampling pump does not describe the analysis with details needed. Please describe what media (on plates) you used for bacteria and mould, what was incubation temperature and time.
Response 4: Added as per comment
“Meanwhile, the Buck Bio-Culture Sampling Pump set at flow rate of 100 liter per minute (LPM) for 2 minutes. Trypticase Soy agar (TSA) for bacterial culture and Malt Extract Agar (MEA) for fungal culture was installed and sealed before transport to laboratory. Trypticase Soy agar (TSA) incubated at 35 0C for 48 hours, meanwhile, Malt Extract Agar (MEA) incubated at 25 0C for 5 days” (Line 165)
Point 5: Air velocity is an important factor for thermal comfort, but it does not give information on air quality. Air exchange rate would be more informative to show the adequancy of ventilation. However, by measuring CO2 you get the indications for this. Did you measure the ventilation rates, and did you investicate the air distribution in the different buildings. Did they differ?
Response 5: During this study, we manage to determine Air Change per Hour (ACH) and compared with standards Air Change Per Hour By ASHRAE Std. 170-2017. Yes, the ACH were different in the areas. However, it was not discussed in this article since the air movement in the most of the areas were below the standard level.
Table 1: Air Change per Hour according to location.
|
Location |
Air Change per Hour (ACH) |
|||
|
HC1 |
HC2 |
HC3 |
HC4 |
|
|
Laboratory (>6*) |
5.28 |
19.58 |
12.52 |
10.16 |
|
Lobby (>4*) |
8.74 |
5.65 |
7.63 |
8.99 |
|
Emergency Room (>6*) |
12.77 |
0.04 |
0 |
0 |
|
Pharmacy (>4*) |
5.47 |
14.31 |
11.07 |
14.71 |
|
Examination Room (>4*) |
0 |
10.49 |
10.8 |
10.99 |
|
*Recommended Air Change Per Hour by ASHRAE Std. 170-2017; HC=Health Clinic |
||||
Point 6: Analysis of the questionnaire results was sufficiently done, although presentation was long as such
Response 6: Thank you for the comments.
Point 7: Table 4. New furniture shows huge number of COR. Is this correct and does it mean that new furniture showed SBS symptoms or not
Response 7: The descriptive analysis resulted all cases with SBS deny any relation expose to new furniture. This part had no value to be analysis during the univariate regression analysis. Considering reviewer comment, this variable removed from the univariate regression analysis.
Point 8: In general use of terminology and language shall be check, there were some minor typos also
Response 8: Edited as per advised
Point 9: SBS is a general term and it does not give tools as such for corrective measures in the buildings if occupants reports different kind of symptoms. Please present in more details based on your findings which symptoms and what reasons were the most important to cause SBS label for the building. Without these details you can't indicate the sources or reasons for symptoms that need to be improved
Response 9: Added as per comment
“Nevertheless, this study highlights inadequate maintenance of Air Handling Unit leading to poor indoor air quality and SBS. The chemical parameters (ozone (O3), carbon monoxide (CO), carbon dioxide (CO2), volatile organic compound (VOC), particulate matters (PM)) resulted within the standard limits. Notwithstanding, physical parameters (temperature (oC) and air movement (AM)) reported unsatisfactory than the threshold level in which concurrent with healthcare workers feedback of poor air quality at their workstation. Subsequently, inefficient indoor air environment enhances microorganism growth in particular area (lobby and emergency room).” (Line 336).
“Failure to maintain efficient AHU invent poor indoor environment that posses health risk for the healthcare workers.” (Line 367).

Reviewer 2 Report
Ecological Study of Sick Building Syndrome among Healthcare 2 Workers at Johor Primary Care Facilities
This is a very interesting study that points out problems associated with SBS in healthcare facilities. The paper is well-written with clear aims, objectives, and methodology. I would add a few comments to further increase the quality of the paper to make it ready for publishing.
In the introduction, authors can point out that SBS is a problem that may be associated with certain climates, contexts, seasonal changes, building type, and activity, space dimension and finishing, ventilation type and efficiency as well as for different occupational types. Accordingly, authors can define the scope, limits, and focus of the study which is focusing on healthcare facilities and different occupancy. This shall prepare readers to understand different perspectives of the problem and acknowledge the authors’ contribution. Authors may refer to some of these references in this concern.
https://www.ncbi.nlm.nih.gov/pmc/articles/PMC7215772/
https://www.tandfonline.com/doi/abs/10.1080/17452007.2022.2060932
https://acp.copernicus.org/articles/15/8217/2015/
https://www.sciencedirect.com/science/article/pii/S0360132322001779
https://www.sciencedirect.com/science/article/pii/S0928493113006450
it is also advised to review different research methods which were used to tackle similar discussions concerning SBS in public buildings and used to validate the selection of your research method and steps adopted.
For each space: Please indicate the type of ventilation system, its schedule of operation, location of inlets and outlets, and indoor materials and furniture setting. This is to note that sometimes, ventilation system and furniture setting can increase SBS in public buildings.
Kindly add the building plans, showing its orientation, and indicate the studied spaces and dimensions and their proximity to the ventilation system
In the analysis, the authors investigated the SBS problem but missed standing on the root causes of SBS for each space based on comparisons following the survey and statistical analysis. This would have helped develop relevant recommendations to alleviate the problem.
CO2: superscript
The conclusion section would benefit from some revision following my previous comments regarding deducting the root cause of the problem and the means to solve it.
Round 2
Reviewer 1 Report
Thank you for the authors for considering the comment dedicated to improve the publication.
Reviewer 2 Report
Dear authors,
Thank you for the modifications carried so far to improve the paper. Nevertheless, the paper needs further revision to be ready for publishing. Kindly find below my review comments:
Use the full term once introduced, e.g. PM
Please add the floor plans to the body of the manuscript…organized in one figure
Authors can delete this part because the scope of the study is on SBS not other maladies.. ‘Contaminated particles can transmit to individuals through inhalation of contaminated aerosols evaporated into the air or fomite exposure. Short-range or long-range exposure depends on the size of the particles. The structure of the micro-organism cell wall is formed by fatty acids and resistance to chemical substances enables micro-organisms to sustain in the environment for long period. Workers are at the risk of short-range airborne transmission and fomite exposure. This is because of work process required them to be in proximity with patient for registration, medical examination and medical procedure [12]. Therefore, the transmission can occur rapidly and lead to an outbreak [13].’
Delete this statement…with a salary of USD 790.54
The conclusion section is very summarized. Kindly revise.
Language check and grammatical mistakes
e.g. According to International Labor Organization (ILO), high risk group such a infants, elderly and individual with chronic disease may potentially at risk of SBS due to indoor air pollutants… missing the verb
e.g. In another study in United State cost approximately USD 5 billion for SBS…revise sentence structure
The selection of the health clinic was based centralized ventilation system (Air Handling Unit (AHU)).. add was based (on).
Biological parameters that measure during the assessments are total counts of bacteria and fungus…revise sentence structure
In this study, only Examination room almost reach the higher standard limit of temperature reading.. check the verb
The metabolic heat which is produced by body.. by the body
Worst in the crowd places, the influence of the crowd may extremely change 296 the indoor air movement …(crowded) and add a full stop to end the sentence
The indoor air movement may affect by ventilation system.. check the verb… add (the) ventilation
on the location of inlet and outlet.. add the
emitted into air …add the
These are study area congested with clients waiting for medical service…missing the verb
. Meanwhile study in Spain…repeated (meanwhile)
Another study among 126 healthcare 356 workers resulted positive relationship was reported…revise sentence…resulted in..or showed
was statistical significant…statistically
Failure to maintain efficient AHU invent…replace invent with a proper verb
The establishing preventive measure….revise sentence structure
Advocation to develop standard operation procedure required to protect workers from SBS at health clinic.. revise sentence structure
